# Interplay between Prokineticins and Histone Demethylase KDM6A in a Murine Model of Bortezomib-Induced Neuropathy

**DOI:** 10.3390/ijms222111913

**Published:** 2021-11-03

**Authors:** Laura Rullo, Silvia Franchi, Giada Amodeo, Francesca Felicia Caputi, Benedetta Verduci, Loredana Maria Losapio, Paola Sacerdote, Patrizia Romualdi, Sanzio Candeletti

**Affiliations:** 1Department of Pharmacy and Biotechnology, Alma Mater Studiorum, University of Bologna, Via Irnerio 48, 40126 Bologna, Italy; laura.rullo3@unibo.it (L.R.); francesca.caputi3@unibo.it (F.F.C.); loredana.losapio@studio.unibo.it (L.M.L.); sanzio.candeletti@unibo.it (S.C.); 2Department of Pharmacological and Biomolecular Sciences, University of Milan, Via Vanvitelli 32, 20129 Milan, Italy; silvia.franchi@unimi.it (S.F.); giada.amodeo@unimi.it (G.A.); benedetta.verduci@studenti.unimi.it (B.V.); paola.sacerdote@unimi.it (P.S.)

**Keywords:** KDM6A, PC1, prokineticins, PPARα, PPARγ, bortezomib, neuropathy, mouse

## Abstract

Chemotherapy-induced neuropathy (CIN) is a major adverse effect associated with many chemotherapeutics, including bortezomib (BTZ). Several mechanisms are involved in CIN, and recently a role has been proposed for prokineticins (PKs), a chemokine family that induces proinflammatory/pro-algogen mediator release and drives the epigenetic control of genes involved in cellular differentiation. The present study evaluated the relationships between epigenetic mechanisms and PKs in a mice model of BTZ-induced painful neuropathy. To this end, spinal cord alterations of histone demethylase KDM6A, nuclear receptors PPARα/PPARγ, PK2, and pro-inflammatory cytokines IL-6 and IL-1β were assessed in neuropathic mice treated with the PK receptors (PKRs) antagonist PC1. BTZ treatment promoted a precocious upregulation of KDM6A, PPARs, and IL-6, and a delayed increase of PK2 and IL-1β. PC1 counteracted allodynia and prevented the increase of PK2 and of IL-1β in BTZ neuropathic mice. The blockade of PKRs signaling also opposed to KDM6A increase and induced an upregulation of PPAR gene transcription. These data showed the involvement of epigenetic modulatory enzymes in spinal tissue phenomena associated with BTZ painful neuropathy and underline a role of PKs in sustaining the increase of proinflammatory cytokines and in exerting an inhibitory control on the expression of PPARs through the regulation of KDM6A gene expression in the spinal cord.

## 1. Introduction

Chemotherapy-induced neuropathy (CIN) is a major adverse effect associated with many cancer therapeutic agents, including bortezomib (BTZ), a first-generation proteasome inhibitor widely used for the treatment of multiple myeloma [1,2,3]. The appearance of this neuropathy strongly affects patient’s life quality and, leading to dose reduction or even to the discontinuation/cessation of chemotherapy, can compromise treatment efficacy with an increase of cancer-related mortality.

Pathological mechanisms of BTZ-induced neuropathy are not yet clearly understood. Similar, however, to what has been suggested for other CIN conditions, involvement of different phenomena, such as oxidative stress, mitochondrial damage, and ion channel dysfunction have been proposed in its development [3,4,5,6,7]. In addition, a role for the immune system in neuroinflammatory processes in this pathological condition has also been indicated [8,9,10]**.** In this regard, we have recently described the involvement of a newly discovered chemokine, the prokineticin 2 (PK2), in the development of BTZ-induced neuropathy [11,12]. The prokineticin family is composed of two proteins: PK1 and PK2 and by the two cognate G protein coupled receptors PKR1 and PKR2, which are widely distributed in pain stations like peripheral nerves, dorsal root ganglia (DRG), and spinal cord [13,14]. The ligand PK2 is recognized as an important player at the crossroad between inflammation and pain [15]. It is known that this chemokine can modulate the immune function, inducing a proinflammatory phenotype and sustaining a proinflammatory condition [16]. Moreover, PK2 sensitizes TRPV1 and TRPA1-expressing nociceptors [17,18] and induces the release of proinflammatory/pro-algogen mediators like cytokines, substance P, and calcitonin gene-related peptide. PK2 is involved in the development of experimental neuropathic pain [14,19,20] of different origin, including BTZ-induced neuropathy [11]. In this regard, we demonstrated a role of PK2 in sustaining a neuroinflammatory condition, triggered by the cytotoxic effect of BTZ in peripheral nerves and DRG. Moreover, the block of PK2 activity with a specific PKRs antagonist could ameliorate sensory hypersensitivity, preserve DRG structure, and reduce the neuroinflammatory condition present in sensory nerves, DRG, and spinal cord of BTZ treated mice [11].

Accumulating evidence suggests that the molecular changes involved in the induction and maintenance of neuropathic pain could also be driven by epigenetic mechanisms. These mechanisms consist of inherited and reversible modifications to nucleotides or chromosomes that are able to alter gene expression without changing DNA sequence. These non-genetic alterations, including DNA methylation and histone modifications, change chromatin state between the transcriptionally accessible euchromatin or inaccessible heterochromatin, thus regulating the transcription of specific genes. As regards histone modifications, while acetylation typically promotes gene transcription, the methylation can either repress or activate this process depending on the lysine (Lys) residue undergoing the modification [21]. Changes in the expression of several epigenetic enzymes carrying out the above-mentioned reactions have been reported in microglia, macrophages, astrocytes, endothelial cells, and neurons after neuronal injuries. Indeed, the activity of methyltransferases and demethylases, as well as of acetylases and deacetylases, is able to alter the promoter state of neuromodulators (e.g., cytokines and chemokines) involved in inflammatory/neuropathic pain and in the sensitization processes [22]. In this respect, it has been described that spinal cord injury causes an increase of the histone demethylase KDM6B in endothelial cells, inducing an increased expression of the cytokine IL-6 by demethylating its promoter [23]. Moreover, KDM6A, another member of the KDM6 family enzymes that specifically demethylates Lys 27 of histone 3 (H3K27me3), epigenetically promotes IL-6 production in macrophages [24]. Interestingly, it has been recently demonstrated that KDM6A gene expression and activity can be upregulated by PK2 via PKR1 in human epicardial stem cells [25]. In the light of the involvement of PK2 in BTZ-induced neuropathy [11], these observations suggest the possible involvement of KDM6A in the epigenetic mechanisms underlying this neuropathic pain condition.

In addition, epigenetic modifications regulated by KDM6A are also reported to affect the gene expression and activity of peroxisome proliferator-activated receptors (PPARs) [25], a class of nuclear receptors known to participate in lipid and glucose metabolism as well as in inflammatory response. In this respect, both PPARα and PPARγ isoforms downregulate the release of pro-inflammatory mediators associated with tissue or nerve injury through the inhibition of proinflammatory signaling pathways, such as NF-κB activation [26]. They are present at key peripheral, spinal, and supraspinal sites involved in pain processing [27], and the increase in PPARs activation or protein expression in the spinal cord has been demonstrated in several animal models of chronic pain [28]. Indeed, several studies suggested that their pharmacological manipulation could represent a promising therapeutic strategy in the control of different types of neuropathic pain [29,30].

Based on this evidence, the present study aimed to evaluate possible relationships among epigenetic mechanisms and prokineticin system involvement in the development and maintenance of bortezomib-induced neuropathy. To this end, alterations in the spinal cord levels of KDM6A and PPARα/ PPARγ, as well as those of PK2 and pro-inflammatory cytokines IL-6 and IL-1β, were assessed in the presence or absence of a specific PKR antagonist, in a model of BTZ-induced neuropathic pain.

## 2. Results

### 2.1. Behavioral Changes and Spinal Cord Biochemical Alterations Induced by BTZ at Day 14

After two weeks of BTZ administration, the presence of allodynia could be ascertained in treated mice that showed a significant decrease of mechanical thresholds compared to control group (4.43 ± 0.12 vs. 8 ± 0.084, *p* < 0.001) (Figure 1).

At the same time point, gene expression analysis showed a significant increase of histone demethylase KDM6A and of the nuclear receptor PPARα mRNA levels in the spinal cord of BTZ-treated-mice (KDM6A: 1.35 ± 0.07 vs. 1.00 ± 0.06, *p* < 0.01; PPARα: 1.45 ± 0.20 vs. 0.97 ± 0.06, *p* < 0.05) (Figure 2A,B). Non-significant changes of PPARγ and PK2 gene expression were observed at this interval (PPARγ: 0.92 ± 0.08 vs. 1.02 ± 0.08, *p* > 0.05; PK2: 1.27 ± 0.11 vs. 1.13 ± 0.01, *p* > 0.05) (Figure 2C,D).

Simultaneously, a significantly higher level of the pro-inflammatory cytokine IL-6 was detected by ELISA in BTZ-treated mice (IL-6: 74.54 ± 12.5 vs. 13.78 ± 1.58, *p* < 0.001), whereas non-significant changes of IL-1β protein were observed at this time point (IL-1β: 49.12 ± 8.85 vs. 43.37 ± 7.02 *p* > 0.05) (Figure 2E,F).

### 2.2. Behavioral Changes and Spinal Cord Biochemical Alterations Induced by BTZ at Day 28 and Effects of PK-Rs Antagonism

As expected, the von Frey test revealed the persistence of mechanical allodynia in BTZ-treated mice. Indeed, a decrease of paw withdrawal threshold (PWT) was observed in these animals that still showed significantly lower PWT than controls (3.2 ± 0.16 vs. 7.5 ± 0.24, *p* < 0.001) (Figure 1) at this time point. However, the daily subcutaneous (s.c.) administration of the PKRs antagonist PC1 from day 14 reverted mechanical allodynia signs. Indeed, at day 28, the PWT value of BTZ + PC1 group appeared significantly higher than BTZ group (7 ± 0.2 vs. 3.2 ± 0.16, *p* < 0.001) and not significantly different from control values (7 ± 0.2 vs. 7.5 ± 0.24, *p* > 0.05) (Figure 1). PC1 alone did not affect paw withdrawal latencies (7.88 ± 0.57 vs. 7.5 ± 0.24, *p* > 0.05).

At this time point, gene expression analysis revealed an increase of KDM6A and PK2 mRNA levels in BTZ-treated mice compared to controls (KDM6A: 1.40 ± 0.09 vs. 1.04 ± 0.07, *p* < 0.05; PK2: 5.43 ± 0.8 vs. 1.13 ± 0.28, *p* < 0.001) (Figure 3A,D), whereas no significant alterations of PPARα and PPARγ gene expression were observed (PPARα: 0.92 ± 0.06 vs. 1.02 ± 0.07, *p* > 0.05; PPARγ: 1.02 ± 0.08 vs. 1.01 ± 0.06, *p* > 0.05) (Figure 3B,C). In addition, BTZ treatment did not cause significant changes of IL-6 protein expression (IL-6: 24.4 ± 2.25 vs. 15.68 ± 2.36, *p* > 0.05) (Figure 3E), whereas a significantly higher level of cytokine IL-1β was detected by ELISA (IL-1β: 62.21 ± 2.64 vs. 39.73 ± 6.81, *p* < 0.01) at this time point (Figure 3F).

As shown in Figure 3A,D, PC1 administration counteracted the BTZ-induced increase of KDM6A and PK2 gene expression. In fact, their mRNA levels in BTZ + PC1 treated mice appeared significantly lower compared to BTZ group (KDM6A: 1.07 ± 0.05 vs. 1.40 ± 0.09, *p* < 0.05; PK2: 1.13 ± 0.16 vs. 5.43 ± 0.8, *p* < 0.001) and did not show any significant differences versus control animals (KDM6A: 1.07 ± 0.05 vs. 1.04 ± 0.07, *p* > 0.05; PK2: 1.13 ± 0.16 vs. 1.13 ± 0.28, *p* > 0.05). Furthermore, PC1 alone did not alter KDM6A or PK2 gene expression compared to control group (KDM6A: 1.10 ± 0.08 vs. 1.04 ± 0.07, *p* > 0.05; PK2: 1.2 ± 0.2 vs. 1.13 ± 0.28, *p* > 0.05) (Figure 3A,D).

As regards PPARs gene expression, a significant increase of both PPARα and PPARγ mRNA levels was observed in BTZ + PC1 treated mice compared to BTZ group (PPARα: 1.38 ± 0.07 vs. 0.92 ± 0.06, *p* < 0.05; PPARγ: 1.42 ± 0.11 vs. 1.02 ± 0.08, *p* < 0.05). Moreover, the PKR antagonist PC1 increased the gene expression of these two nuclear receptors compared to control mice also when administered alone (PPARα: 1.65 ± 0.15 vs. 1.02 ± 0.07, *p* < 0.001; PPARγ: 1.44 ± 0.09 vs. 1.01 ± 0.06, *p* < 0.05) (Figure 3B,C).

Concerning cytokines protein levels, PC1 did not affect IL-6 levels either when administered with BTZ (IL-6: 19.54 ± 4.07 vs. 24.4 ± 2.25, *p* > 0.05) or when injected alone (IL-6: 14.62 ± 2.69 vs. 15.68 ± 2.36, *p* > 0.05) (Figure 3E). Instead, this PKRs antagonist was able to counteract the BTZ-induced increase of IL-1β (IL-1β: 43.25 ± 1.96 vs. 62.21 ± 2.64, *p* < 0.05) without affecting the cytokine levels by itself (IL-1β: 38.51 ± 4.84 vs. 39.73 ± 6.81, *p* > 0.05) (Figure 3F).

## 3. Discussion

Neuroinflammation represents one of the main mechanisms underlying BTZ-induced neuropathic pain, and chemokines are emerging as important mediators in this pathway. In particular, among chemokines, we recently suggested a role of the prokineticins- PKs [11,12,31], demonstrating their role in sustaining neuroinflammation in peripheral nerves and DRG. We also showed that PK system activation in peripheral nervous system (PNS) is important for promoting spinal cord activation and central sensitization [11,31]. In the light of the ability of PKs to affect the expression and activity of epigenetic enzymes involved in the regulation of inflammatory processes [23,24,25], and to better understand how this chemokine family could promote BTZ-induced neuropathy, we evaluated the influence of PK signaling modulation upon consideration of different biochemical parameters during the development of this painful condition.

After two weeks of BTZ administration, when allodynia signs were significantly developed, we observed a significant increase of the histone demethylase KDM6A gene expression in the spinal cord of pain suffering mice. This result is in agreement with the relevant role suggested for demethylating enzymes to ensure rapid inflammatory responses. Indeed, the rapid erasure of repressive histone marks by KDM6A is considered as essential for NF-κB-dependent gene regulation [32]. The observed increase of mRNA levels for this enzyme is accompanied by a significant enhancement of IL-6 protein levels, consistent with the reported KDM6A ability to epigenetically promote the production of this cytokine [24] known to participate in the dysregulation of interendothelial junctions and favouring leukocyte adhesion and migration [33]. In that regard, loosening of the blood–brain barrier has been highlighted as an important event involved in chemotherapy-induced neurotoxicity [34].

At the same time point, results indicated that BTZ caused the significant increase of PPARα gene expression, whereas no changes were detected for PPARγ mRNA levels. The early increase of PPARα in other CIN conditions has been observed [35], thus suggesting that this PPAR isoform could represent a relevant regulatory first line response attempting to dampen inflammatory conditions, involving phenomena such as endothelial loosening and leukocyte migration [36].

The investigated parameters showed a different alteration pattern after four weeks of BTZ administration protocol. Indeed, at day 28, the still significantly elevated KDM6A gene expression was accompanied by the significant increase of PK2 mRNA levels and of IL-1β protein, while IL-6 protein levels were no more distinguishable from control values. At the same time, PPARα levels returned to basal values, thus underlining their main involvement in the early anti-inflammatory responses [35].

The delayed upregulation of PK2 in spinal cord confirms what we previously reported in BTZ treated animals, but also in neuropathy induced by the chemotherapeutic vincristine [11,31]. In contrast, significant upregulation of PK2 in spinal cord in neuropathic mice is precociously present in other models of painful neuropathy, such as chronic constriction injury, spinal nerve ligation, and diabetic peripheral neuropathy [14,19,20]. These results suggest that in CIN, PK2 is particularly involved in sustaining and maintaining neuropathic pain rather than in its onset. PK2 may act in an autocrine or paracrine way, sustaining a neuroinflammatory loop that exacerbates the neuronal damage and drives a progressive sensitization in spinal cord. PK2 up-regulation is induced through STAT3 activation that binds the enhancer site of its promoter [37,38,39]. STAT3 activation by G-CSF, IL-6, and IL-1β signaling was recently demonstrated in DRG, spinal cord neurons and astrocytes [13,40]. PK2 may contribute to microgliosis, astrocytosis, and production of proinflammatory cytokines, such as IL-1β and IL-6, which in turn stimulate astrocytes and neurons to induce further PK2 expression suggesting the presence of a feed forward loop. It was also suggested that the overexpression of PK2 and PKR2 on activated astrocytes can act as an astrocytic-autocrine-growth factor [13]. In support of PK2 feed forward loop, Nebigil’s group [41] demonstrated that PKR1 signaling in cardiomyocyte upregulates PK2, which acts as a paracrine factor for triggering the proliferation/differentiation of epicardial-derived progenitor cells (EDPC) [41]. As a consequence, and in line with previous studies [11], the repeated administration of the PKRs antagonist PC1 was able to counteract allodynia signs, causing a significant elevation of mechanical thresholds of BTZ-treated mice up to control values at the 28th day. In fact, the prolonged pharmacological blocking of these receptors, together with the reduced availability of the agonist, blunted PNS neuroinflammation, preserved DRG structure with consequent reduction of sensitization of the pathway responsible for allodynia [11,14]. Interestingly, the blockade of PKRs signaling also affected some of the BTZ-induced biochemical alterations above-mentioned, thus suggesting PK system involvement in their regulation.

The ability of PC1 to counteract both PK2 and KDM6A gene expression levels caused by BTZ at day 28 highlights that prokineticins can control the expression and activity of some epigenetic enzymes in vivo, according to what was reported in cell cultures [25].

Moreover, the capacity of PC1 to prevent BTZ-induced increase of some proinflammatory cytokines, such as IL-1β, suggests that PK system might epigenetically regulate inflammatory processes involved in pain maintenance. In this regard, previous studies showed the KDM6A’s ability to regulate H3k27me3 demethylation at the IL-1β gene promoter [42].

Of note, blocking the binding of PK2 to its receptors induced a significant upregulation of both PPARs isoforms. In fact, PC1 antagonist was able to increase PPARs mRNA levels in the absence of neuropathy and this effect was maintained also in the presence of BTZ-induced neuropathic pain. These results could suggest the existence of a tonic inhibitory control exerted by PK2 on the expression of PPARs. In this view, the blockade of PKRs in BTZ-treated mice would maintain and prolong the upregulation of PPARα gene expression that, as shown here, was significantly higher than controls at 14 days after BTZ mice and spontaneously returned to basal levels at day 28.

Moreover, when PK2 signaling was abolished by PC1, also PPARγ, which was not upregulated in BTZ mice at both intervals of investigation, was overexpressed.

In the light of KDM6A’s ability to repress PPARγ expression and activity [25] and given the link between PK2 and KDM6A here shown, it can be hypothesized that the PK2 modulation of PPAR gene expression could be epigenetically mediated by histone demethylases in the spinal cord of BTZ-induced neuropathy suffering mice.

In previous work, it has been demonstrated that PK2 is also able to decrease the production of the anti-inflammatory cytokine IL-10, both in peripheral immune cells and in spinal cord glial cells [12,15,16,20]. The inhibitory role of PK2 on PPARs that, when activated, dampens inflammatory responses and relief pain, further sustains a relevant role of PK2 as pro-inflammatory mediator, since it stimulates proinflammatory and pronociceptive factors but also blunts the endogenous anti-inflammatory responses [13].

In conclusion, these data showed the involvement of epigenetic modulatory enzymes in spinal tissue phenomena associated to BTZ painful neuropathy, and underline a role of the PK system in sustaining the increase of specific proinflammatory cytokines through the regulation of the histone demethylase enzyme KDM6A gene expression.

## 4. Materials and Methods

### 4.1. Animals

Male C57BL/6J nine-week-old mice (Charles River Laboratories, Calco, Italy) were used. Before starting experiments, mice were acclimatized for seven days at 22 ± 1 °C room temperature and 55 ± 10% humidity with 12-h dark/light cycle and food and water *ad libitum*. Mice were handled daily by exposure to a passive hand and tickling.

All procedures comply with ARRIVE guidelines and were carried out in accordance with 2010/63/EU directive for animal experiments and with International Association for the Study of Pain and European Community (E.C.L358/118/12/86) guidelines and were approved by the Animal Care and Use Committee of the Italian Ministry of Health (permission number 709-2016, 07/22/2016 to SF). In accordance with 3R principles, all efforts were made to reduce the number of animals used and their sufferance.

### 4.2. Painful Neuropathy Induction and Therapeutic Treatment with the PKR Antagonist, PC1

Bortezomib (BTZ, LC Laboratories; Woburn, MA, USA) was intraperitoneally (i.p.) injected at the dose of 0.4 mg/kg three times a week (every Monday, Wednesday, Friday) for a total of four consecutive weeks. The PKRs antagonist PC1 [11,43], a triazine–guanidine compound, was subcutaneously administered to BTZ mice (BTZ + PC1) or to control mice (PC1) at the dose of 150 μg/kg twice a day for 14 consecutive days, starting from day 14 until the end of the BTZ protocol (day 28) [11].

### 4.3. Von Frey Test: Mechanical Allodynia

Mechanical allodynia was assessed using a Dynamic plantar Aesthesiometer (Ugo Basile, Gemonio, Italy), evaluating the mechanical touch sensitivity to a blunt probe (Von Frey filament, 0.5 mm diameter, ranging up to 10 g in 10 s) applied to the central plantar surface of the hind-paw. Response to mechanical stimuli PWT was expresses in grams (g) [31]. Three different measurements for each paw were recorded and the mean was calculated. Behavioral evaluations were performed at day 14 and 28.

### 4.4. Tissue Collection

At day 14, before starting PC1 treatment (groups: CTR and BTZ) and at the end of the BTZ schedule (day 28), corresponding to 14 days of PC1 treatment (groups: CTR, BTZ, BTZ+ PC1, and PC1), mice were sacrificed by decapitation. For each mouse, spinal cord (L4-L5) was collected, immediately frozen in nitrogen, and stored at −80 °C for successive evaluations.

### 4.5. RNA Extraction and Gene Expression Analysis by Real-Time qPCR

Total RNA was extracted from the spinal cord samples using the Quick™ DNA/RNA MiniPrep (cat. # D7001, Zymo Research, Orange, CA, USA) according to the manufacturer’s instructions. RNA integrity was checked by 1% agarose gel electrophoresis and concentrations were measured by using a Nanodrop 1000 system spectrophotometer (Thermo Fisher Scientific, Waltham, MA, USA). RNA samples with 260/280 nm absorbance ratio > 1.8 and < 2.0 were subsequently reverse-transcribed with the GeneAmp RNA PCR kit (Life Technologies, Carlsbad, CA, USA). Relative abundance of each mRNA of interest was assessed by real-time qRT-PCR using the Sybr Green gene expression Master Mix (Life Technologies, Carlsbad, CA, USA) in a Step One Real-Time PCR System (Life Technologies, Carlsbad, CA, USA) as previously described [44]. The primers used for PCR amplification in SYBR Green PCR MasterMix were designed using Primer 3 and are here reported: GAPDH Forward 5′-AACTTTGGCATTGTGGAAGG-3′; GAPDH Reverse 5′-ACACATTGGGGGTAGGAACA-3′; KDM6A Forward 5′-TTTGGTCTACTTCCATTA-3′; KDM6A Reverse 5′-AAGCCCAAGTCGTAAATGAATTTC-3′; PPARα Forward 5′-AGGGTTGAGCTCAGTCAGGA-3′; PPARα Reverse 5′-GGTCACCTACGAGTGGCATT-3′; PPARγ Forward 5′-GGAAGACCACTCGCATTCCTT-3′; PPARγ Reverse 5′-GTAATCAGCAACCATTGGGTCA-3′.

For PK2 mRNA evaluation, Taqman Gene expression assays (PK2: Mm01182450_g1; GAPDH, Mm99999915_g1; Thermo Fisher Scientific, Waltham, MA, USA) and Luna^®^ Universal Probe qPCR Master Mix (BioLabs, London, UK) were used.

Each sample was run in triplicate and data were normalized to those of the endogenous reference gene GAPDH. Relative expression of different gene transcripts was calculated by the Delta-Delta Ct (DDCt) method and converted to relative expression ratio (2−DDCt) for statistical analysis [45].

### 4.6. Cytokine Protein Content Measurement

Spinal cord samples were homogenized by means of Ultra Turrax homogenizer in a volume of 500 μL of sample buffer (phosphate saline buffer plus protease inhibitor cocktail (Roche, Monza, Italy), added with EDTA (SigmaAldrich, Milan, Italy), and centrifuged [46] for 15 min at 13,000 rpm at 4 °C. Supernatants were collected and used for total protein content determination (Lowry method) and for measuring cytokine levels.

Protein levels of IL-6 and IL-1β were determined by enzyme-linked immunosorbent assay (ELISA) using an ultrasensitive ELISA kit according to the manufacturer’s instructions (eBioscience, San Diego, CA, USA). Sensitivity: 4 pg/mL and 8 pg/mL for IL-6 and IL-1β, respectively.

### 4.7. Data Analysis

Behavioral and biochemical data were evaluated by Shapiro–Wilk tests to confirm the normality of the distribution and by Grubb’s test to identify outliers. Statistical analysis was performed at day 14 by using t-test and at day 28 by one-way ANOVA followed by Bonferroni’s test for multiple comparisons. All statistical analyses were performed using GraphPad 9 software (San Diego, CA, USA). Results are expressed as mean ± standard error of the mean (SEM) (*n* = 6 animals/group). Differences were considered significant at *p* < 0.05.

## Figures and Tables

**Figure 1 ijms-22-11913-f001:**
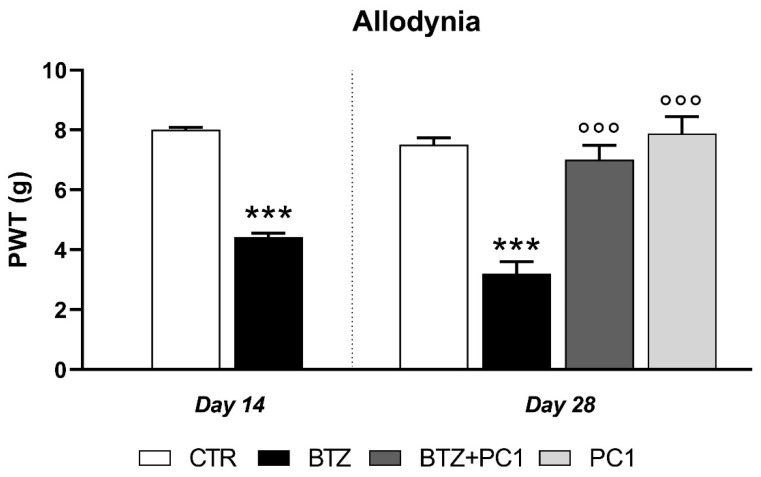
Mechanical allodynia induced by BTZ treatment (0.4 mg/kg, 3 times/week for two (Day 14), or four weeks (Day 28)) and effect of PC1. PC1 (150 μg/kg s.c., twice daily) was administered alone or together with BTZ for 14 days, starting from day 14 (established mechanical allodynia) until day 28. Data represent the mean ± SEM of paw withdrawal threshold (PWT, in grams) from six mice/group. (*** *p* < 0.001 vs. CTR; °°° *p* < 0.001 vs. BTZ; analyzed by *t*-test (Day 14) or one-way ANOVA followed by Bonferroni’s multiple comparison test (Day 28)).

**Figure 2 ijms-22-11913-f002:**
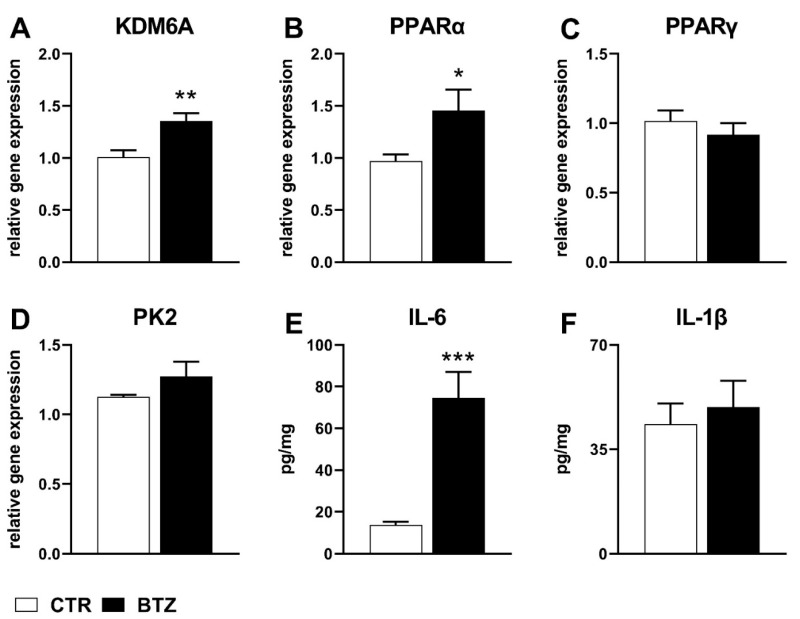
Spinal cord biochemical alterations assessed at the day 14 after BTZ treatment (0.4 mg/kg, 3 times/week for two weeks). (**A**–**D**) mRNA levels determined by qPCR for KDM6A, PPARα, PPARγ, and PK2. (**E**,**F**) protein levels determined by ELISA for IL-6 and IL-1β. Data are expressed as mean ± SEM of six mice/group (* *p* < 0.05; ** *p* < 0.01; *** *p* < 0.001 vs. CTR; analyzed by *t*-test).

**Figure 3 ijms-22-11913-f003:**
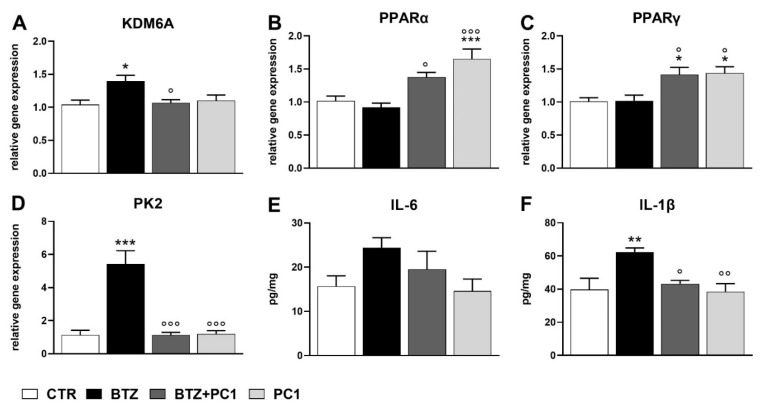
Spinal cord biochemical alterations assessed at the day 28 after BTZ treatment (0.4 mg/kg, 3 times/week for 4 weeks) and effect of PC1. PC1 (150 μg/kg s.c., twice daily) was administered alone or together with BTZ for 14 days, starting from day 14 until day 28. (**A**–**D**) mRNA levels determined by qPCR for KDM6A, PPARα, PPARγ, and PK2. (**E**,**F**) protein levels determined by ELISA for IL-6 and IL-1β. Data are expressed as mean ± SEM of six mice/group (* *p* < 0.05; ** *p* < 0.01; *** *p* < 0.001 vs. CTR; ° *p* < 0.05; °° *p* < 0.01; °°° *p* < 0.001 vs. BTZ; analyzed by one-way ANOVA followed by Bonferroni’s multiple comparison test).

## Data Availability

Data supporting the findings of this study are available upon reasonable request.

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
