# Peer review of "Interplay between Prokineticins and Histone Demethylase KDM6A in a Murine Model of Bortezomib-Induced Neuropathy"

_ijms, 2021, doi:10.3390/ijms222111913_

Round 1
Reviewer 1 Report
The study of Rullo et al. evaluates the relationship between epigenetic mechanisms and PKs in a mice model of BTZ-induced painful neuropathy (BINP). However, neuropathy affects the degeneration of peripheral nerve fibers, the authors described changes in gene expression only in the spinal cord. Therefore, it is not possible to determine the effect of PC1 on morphological changes and protein expression in the peripheral nerves, they probably lead to an improvement of allodynia in CIPN after therapy. This point should be augmented in the discussion.
Major points:
- The general underlying pathology of CIPN is axonal degeneration. The evaluation of axons (axon numbers, axonal atrophy) in a peripheral nerve +/- PC1 and state of myelination should be determined.
- The increased/decreased inflammatory response (enhanced macrophage infiltration and proliferation) in the peripheral nerves with immunofluorescence methods or protein expression analysis (WB) should be also investigated.
- The inflammatory profile is lacking of TNFalpha - a critical regulator of immune function.
- Generally, there is no connection between gene regulation and protein expression in the peripheral nerves!
- What does it mean? “BTZ promoted upregulation of PK2 and IL-1 at later time“ - please clarify this sentence.
Author Response
Please see the attachment below.

Reviewer 2 Report
Comments are attached

Author Response
Please see the attachment below.

Round 2
Reviewer 1 Report
please add TNF results in the Figures, it may be helpful in interpreting the specific inflamatory profile of this neuropathy